# Text Fact Transfer

**Nishant Balepur     Jie Huang     Kevin Chen-Chuan Chang**

University of Illinois at Urbana-Champaign, USA

{balepur2, jeffhj, kcchang}@illinois.edu

## Abstract

Text style transfer is a prominent task that aims to control the style of text without inherently changing its factual content. To cover more text modification applications, such as adapting past news for current events and repurposing educational materials, we propose the task of text fact transfer, which seeks to transfer the factual content of a source text between topics without modifying its style. We find that existing language models struggle with text fact transfer, due to their inability to preserve the specificity and phrasing of the source text, and tendency to hallucinate errors. To address these issues, we design ModQGA, a framework that minimally modifies a source text with a novel combination of end-to-end question generation and specificity-aware question answering. Through experiments on four existing datasets adapted for text fact transfer, we show that ModQGA can accurately transfer factual content without sacrificing the style of the source text.[1]

## 1   Introduction

Text style transfer aims to *control the stylistic attributes* of text, such as sentiment or formality, without affecting its factual content (Jin et al., 2022; Hu et al., 2022). This task has several applications, including personalizing dialogue agents (Rao and Tetreault, 2018; Zheng et al., 2020), increasing persuasiveness in marketing or news (Jin et al., 2020; Moorjani et al., 2022), or simplifying educational resources (Wang et al., 2019; Cao et al., 2020).

While text style transfer models can adeptly alter stylistic elements, they do not address all text modification needs, especially those centered on factual modifications. Specifically, there exist several applications that require the *transfer of factual content between topics* without altering style, such as adapting past news articles for current events

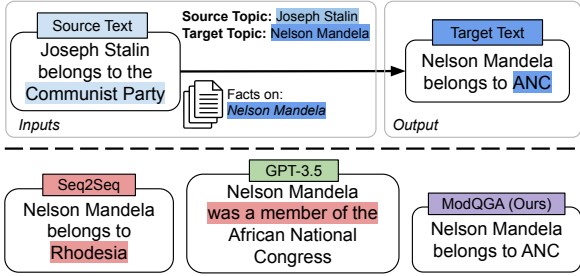

Figure 1: (Top) Overview of text fact transfer. (Bottom) Outputs from the seq2seq LED language model, 0-shot GPT-3.5, and 0-shot ModQGA (ours) on text fact transfer. Red highlighted text indicates factual inaccuracies or failure to match the style of the source text.

(Graefe, 2016) and repurposing educational materials for new subjects (Kaldoudi et al., 2011), which are outside the scope of text style transfer. Further, studying methods to transfer facts while preserving style could be useful for augmenting datasets, i.e., expanding training sets with new, factual training examples in a similar style (Amin-Nejad et al., 2020; Bayer et al., 2023), or evaluating the factual accuracy of text generation models (Celikyilmaz et al., 2020; Ji et al., 2023).

To address these needs, we propose the task of **text fact transfer**, which aims to modify the factual content of a source text while preserving its style. We define factual content as topic-specific entities that convey knowledge and style as *how* the factual content is phrased and organized, as well as its level of specificity[2]. As shown in Figure 1 (top), given as inputs a source text, source topic, target topic, and corpus of facts for the target topic, we seek to generate a target text that matches the style of the source text and contains factual content specific to the target topic. Thus, while text style transfer aims to modify *subjective, stylistic* aspects of text, text

---

[1]Code is available at https://github.com/nbalepur/text-fact-transfer.

[2]Depending on the setting, this definition of style may need to be modified. For example, in educational repurposing, it may be infeasible to keep the phrasing consistent, as different subjects may need to be discussed and phrased differently.

fact transfer controls the *objective, factual* content.

One approach for text fact transfer (on parallel corpora) is to train/prompt seq2seq or large LMs (Lewis et al., 2020a; Brown et al., 2020). However, there are two inherent challenges to text fact transfer that cannot be overcome by directly applying these models. **First**, the generated text must not deviate from the wording of the source text, but LMs may not always succeed in this regard (Balepur et al., 2023). For example in Figure 1, GPT-3.5 states that Nelson Mandela "was a member of" the ANC, which is inconsistent with the phrasing of "belongs to" present in the source text.

**Second**, along with being accurate, the factual content must align with the specificity of the source text to best maintain its style. However, LMs have been shown to hallucinate (Ji et al., 2023) and struggle to control the specificity of their outputs (Huang et al., 2022). For example, as seen in Figure 1, the seq2seq LM states that "Nelson Mandela belongs to Rhodesia." Although the leader has some links to Rhodesia, it is inaccurate to state that he belongs there. Further, the source text contains the political party of Joseph Stalin (i.e., Communist Party), so the target text should contain a political party (i.e., ANC) rather than a country, which is less specific.

Further, these challenges become more complex if supervised text fact transfer is infeasible. For example, when adapting past news for current events or augmenting datasets, it could take ample time and effort to construct parallel corpora and train a supervised model. In such cases, **0-shot** models, while harder to develop, are preferred, as they can adapt to domains without extra training. Hence, we must study 0-shot and supervised text fact transfer models to ensure adaptability in downstream tasks.

To address these challenges of text fact transfer, we extend the concept of minimal alterations for text style transfer proposed by Li et al. (2018) and seek to execute the two-step process of: (1) locating factual entities in the source text; and (2) solely transferring these entities between topics. To perform **step one**, we note that factual entities are inherently question-driven, and thus any entity in the source text that must be transferred can answer a question. For example in Figure 1, the factual entity "Communist Party" answers the question "What is Joseph Stalin's party?". To perform **step two**, we find that transferring entities between topics is challenging, but transferring *questions* that can retrieve said entities is simple. For example, transferring "Communist Party" to "ANC" directly is difficult, but we can easily transfer "What is Joseph Stalin's party?" to "What is Nelson Mandela's party?" by replacing the source topic (Joseph Stalin) with the target topic (Nelson Mandela), returning a question that can be used to retrieve the entity "ANC."

Exploiting these findings, we design **ModQGA**, a model that minimally modifies the source text with a combination of **Q**uestion **G**eneration (QG) and **A**nswering (QA). As shown in Figure 2, ModQGA first uses end-to-end QG to jointly produce entities from the source text and questions that can be answered by said entities. Next, these questions are transferred to pertain to the target topic. ModQGA then uses specificity-aware QA to retrieve an answer from the corpus for each transferred question, while matching the specificity of the source text entities. Finally, these answers are filled into the source text. Solely modifying factual entities allows for the preservation of the phrasing of the source text, while the focused approach of transferring entities with specificity-aware QA promotes factuality and matched specificity, as shown in Figure 1. Further, we can train the QG and QA models of ModQGA on external QA datasets (Rajpurkar et al., 2016), resulting in a 0-shot model that can be applied to diverse domains without extra training.

We showcase the strength of ModQGA for text fact transfer by creating four parallel corpora from existing datasets, spanning expository text generation (Balepur et al., 2023) and relationship triples (Elsahar et al., 2018; Gardent et al., 2017). Hence, our initial study of text fact transfer focuses on the adaptation of expository texts and relationship triples, leaving applications such as repurposing news articles and dataset augmentation for future research. Using these datasets, we design a 0-shot and supervised version of ModQGA and in our experiments, find that both models outperform their respective baselines in style preservation and factuality on a majority of datasets.

Our contributions can be summarized as follows:

**1)** We propose the task of text fact transfer, which aims to alter factual content while preserving style.
**2)** To solve our task, we design ModQGA, which minimally modifies a source text with an ensemble of end-to-end QG and specificity-aware QA. We qualitatively assess the latter, which shows at least some ability to control the specificity of its answer.
**3)** We adapt four datasets for text fact transfer.
**4)** Through experiments on our four datasets, we

demonstrate that ModQGA generates factual text that is stylistically consistent with the source text.

## 2 Related Work

### 2.1 Text Style Transfer

Text style transfer aims to modify the style of text without inherently affecting its content (Fu et al., 2018; Jin et al., 2022; Hu et al., 2022). The concept of style can take many forms, including formality (Wang et al., 2019; Zhang et al., 2020), sentiment (Prabhumoye et al., 2018; Yang et al., 2018), and authorship (Xu et al., 2012). Text fact transfer is the counterpart to text style transfer, as we focus on transferring the factual content of text between topics without affecting its underlying style. Hence, our task emphasizes generating new, factual text, which is not the main focus of style transfer tasks.

Several methods have been developed for text style transfer, such as training neural models on parallel corpora (Rao and Tetreault, 2018; Xu et al., 2019), latently disentangling content and style (Hu et al., 2017; Shen et al., 2017), or prototype editing (Li et al., 2018; Sudhakar et al., 2019; Abu Sheikha and Inkpen, 2011). ModQGA is most similar to the Delete-Retrieve-Generate model (Li et al., 2018), which extracts attribute markers, transfers attributes across styles, and generates an output. We apply a similar technique for text fact transfer, but notably use a novel combination of end-to-end question generation and specificity-aware question answering, which has not been explored in prior work.

### 2.2 Stylistic Exemplars

Recent work has studied models that leverage stylistic exemplars to guide stylistic choices in text generation (Cao et al., 2018; Wei et al., 2020). Such exemplars improve the fluency of seq2seq models in various tasks, including summarization (Dou et al., 2021; An et al., 2021), machine translation (Shang et al., 2021; Nguyen et al., 2022), dialogue generation (Zheng et al., 2020; Wang et al., 2021), and question answering (Wang et al., 2022).

More relevant to text fact transfer are tasks that require *strictly* adhering to the style of an exemplar. Chen et al. (2019) propose the task of controllable paraphrase generation, which aims to combine the semantics from one sentence and the syntax from a second sentence. Lin et al. (2020) introduce "style imitation" and perform data-to-text generation while strictly maintaining the style of an ex-

emplar. Apart from a lack of focus on transferring factual content, these works differ from our task as they do not leverage a factual corpus.

The task most similar to ours is expository text generation (ETG) (Balepur et al., 2023), which seeks to generate factual text from a corpus in a consistent style. However, ETG dictates that this style is learned from examples of outputs in the same domain, while text fact transfer adheres to the style of a single source text. Hence, an ETG model is domain-specific, while a single text fact transfer model (e.g., 0-shot ModQGA) could be used in several domains. Further, the IRP model proposed by Balepur et al. (2023) for ETG combines content planning, retrieval, and rephrasing, while ModQGA modifies a source text with question generation and answering, and our model tackles the additional problem of controlling specificity (§3.3).

### 2.3 Analogy Completion

The concept of transferring entities between topics is similar to analogy completion (Ushio et al., 2021; Bhavya et al., 2022; Chen et al., 2022), which aims to select a word that parallels an input query-word pair (e.g., "Paris:France, Lima:[MASK]"). While analogy completion could be used for factual entity transfer, this is only one aspect of text fact transfer. Further, our task is fundamentally a text generation task, while analogy completion is typically used to assess how models internally capture relations.

## 3 Methodology

Given a source text $\mathcal{D}_s$, source topic $t_s$, and target topic $t_t$, text fact transfer aims to produce a target text $\mathcal{D}_t$ that matches the style of $\mathcal{D}_s$ and modifies the entities related to the source topic $t_s$ with entities related to the target topic $t_t$. To serve as ground truth information for $t_t$, we also provide a corpus of factual sentences $\mathcal{C}$ related to the target topic $t_t$.

As illustrated in Figure 2, the backbone of Mod-QGA consists of two key modules: (i) An end-to-end question generator $p(\mathcal{Q}_s|\mathcal{D}_s, t_s)$ that produces question/entity pairs $(q, e) \in \mathcal{Q}_s$, where each $q$ can be answered by $e$ using the source text $\mathcal{D}_s$; and (ii) A specificity-aware question answering model $p(\langle a_i, a_j \rangle|c, q, e)$ that extracts an answer span $\langle a_i, a_j \rangle$ from the context $c$ (where $c \subseteq \mathcal{C}$), which answers question $q$ and matches the specificity of the entity $e$. After training these models, ModQGA performs text fact transfer via: 1) end-to-end question generation with $p(\mathcal{Q}_s|\mathcal{D}_s, t_s)$; 2)

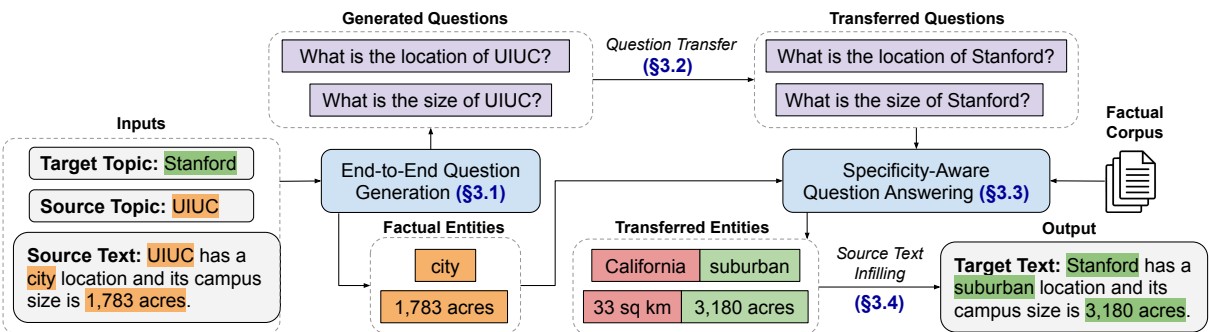

Figure 2: Overview of ModQGA. First, ModQGA jointly generates questions and factual entities with an end-to-end question generator. These questions are then transferred to pertain to the target topic. Next, ModQGA performs specificity-aware question answering to retrieve answers from the factual source that match the level of specificity of the source entity (correct/incorrect matches in green/red). Last, ModQGA infills the source text with these answers.

question transferring; 3) question answering with $p(\langle a_i, a_j\rangle | c, q, e)$; and 4) source text infilling. We will describe each of these steps followed by how they are combined for the full ModQGA model.

## 3.1 End-to-End Question Generation

Our approach to text fact transfer is rooted in the observation that all entities in the source text that need to be transferred can be viewed as an answer to a question. For example, given the source text "Ibuprofen is used to relieve pain," transferring between the topics Ibuprofen and Melatonin may result in the text "Melatonin is used to promote sleep." The part of the source text that needs to be transferred (apart from the known transfer of "Ibuprofen" to "Melatonin") is "to relieve pain," which can answer the question "Why is Ibuprofen used?". This question-answer paradigm helps us guide the modification process in text fact transfer.

Hence, to identify entities that need to be transferred and the questions that can be answered by said entities, we train an end-to-end question generation model that jointly generates entities and their questions from a context. To do so, we leverage the SQuAD-V2 dataset (Rajpurkar et al., 2016). We train BART-large (Lewis et al., 2020a) to minimize the loss $\lambda_{qg}$ of token prediction of question $q$ and answer (entity) $e$, surrounded by `<|question|>` and `<|answer|>` tokens (represented as $\langle q \cdot e\rangle$), conditioned on the context $c$ and topic $t$:

$$\lambda_{qg} = -\sum_{i=1}^{|\langle q \cdot e\rangle|} \log p(\langle q \cdot e\rangle_i | c, t, \langle q \cdot e\rangle_1, ..., \langle q \cdot e\rangle_{i-1}). \quad (1)$$

If a generated question $q$ contains the source topic $t_s$, $q$ can be simply transferred to the target topic $t_t$ by replacing $t_s$ with $t_t$. To elicit this desirable

property, we only keep SQuAD entries where the topic is a substring of the question.[3] Thus, all training questions contain the topic $t_t$, teaching the model to produce $t_t$ in the output during inference.

To ensure all factual entities in the source text are detected, we use nucleus sampling to generate $n$ question/entity pairs $\mathcal{Q}_s = \{(q_j, e_j)\}_{j=1}^n$ with each sentence of the source text $\mathcal{D}_s$ as the context $c$ and source topic $t_s$ as the topic $t$. Thus, each unique factual entity from the source text may be mapped to multiple questions, which are ensembled by the specificity-aware question answering model (§3.3).

As a final post-processing step, we discard pairs in $\mathcal{Q}_s$ with an entity that does not appear in the source text, as this entity is hallucinated. Further, if an entity is a substring of another entity in $\mathcal{Q}_s$, we discard the substring (shorter) entity.

## 3.2 Question Transferring

Each question $q_j$ found in the question/entity pairs $(q_j, e_j) \in \mathcal{Q}_s$ pertain to the source topic $t_s$, but we require a transferred question $q'_j$ that pertains to the target topic $t_t$. Since $q_j$ will contain the substring $t_s$, we can simply replace $t_s$ with $t_t$ to obtain $q'_t$.

Through testing on our validation sets, we also find that *generic* queries can outperform *specific* queries when retrieving contexts for question answering. For instance, we find that the generic query "What is the hub of the airport?" outperforms the specific query "What is the hub of Cathay Pacific Airport?". We find this occurs because the inclusion of the topic $t_t$ in the query distracts the retriever when searching $\mathcal{C}$ for contexts in QA, as it is more biased towards facts that contain the to-

---

[3]We found that performing lexically constrained token decoding (Hokamp and Liu, 2017) with topic $t_t$ led to a similar outcome, but this resulted in higher GPU memory usage.

kens in $t_t$, even when said facts are not relevant to the query.[4] We show the benefit of generic queries experimentally with ablation studies (§5.3).

To obtain a generic question $q_j''$, we take the intersecting tokens of $q_j$ and $q_j'$, which eliminates the topic-specific tokens found in $t_s$ and $t_t$. Combining the specific and generic questions, we obtain a set of transferred questions and their corresponding source entities $\mathcal{Q}_t = \{(q_j', e_j)\}_{j=1}^n \cup \{(q_j'', e_j)\}_{j=1}^n$.

## 3.3 Specificity-Aware Question Answering

After creating the transferred question-entity pairs $\mathcal{Q}_t$, we faithfully answer each transferred question by retrieving a context from the factual source $\mathcal{C}$ followed by extractive question answering (QA). However, an off-the-shelf QA model cannot be used for our task, as it fails to consider the specificity of the answer we require (Huang et al., 2022). To extract transferred entities that are stylistically aligned with the source text, we seek answers with the same level of specificity as the entities they are replacing. For example, at one step in ModQGA, we may obtain the question "Where is Stanford located?" derived from the source entity "rural". While "California," "Palo Alto," and "suburban" are all valid answers, "suburban" is the best choice, as it shares the same level of specificity as "rural," and thus best matches the style of the source text.

To create a dataset with these specifications, we again modify the SQuAD-V2 dataset (Rajpurkar et al., 2016). The dataset already provides questions, contexts, and answer spans, but we still require guidance to match the specificity levels of the answers. We find that one way to obtain specificity guidance of an answer is through the skip-gram assumption (Mikolov et al., 2013)—similar words are discussed in similar contexts. For example, we intuit that because "rural" and "suburban" are used in the same context (e.g., "the location is [suburban/rural]"), they have similar specificity levels. Hence, we obtain specificity guidance for each answer in the SQuAD dataset by replacing every word in the answer with a random top-20 skip-gram synonym via fastText embeddings (Joulin et al., 2017).

We use BERT-large (Devlin et al., 2018) to train our specificity-aware QA model $p(\langle a_i, a_j \rangle | c, q, e)$. We minimize $\lambda_{qa}$, the sum of $\lambda_i$, the cross-entropy loss of the predicted start index $a_i$ and $\lambda_j$, the loss of the predicted end index $a_j$, conditioned on the

---

[4]We note that the specific query *can* find the facts with top-$k$ retrieval for large $k$, but our QA model is trained to use fewer contexts ($k = 5$), hence our need for generic queries.

context $c$, question $q$, and specificity guidance $e$:

$$\lambda_i = -\sum_{z=1}^N \log p(a_i|c,q,e)\mathbb{I}(z=i), \quad (2)$$

$$\lambda_j = -\sum_{z=1}^N \log p(a_j|c,q,e)\mathbb{I}(z=j), \quad (3)$$

$$\lambda_{qa} = \lambda_i + \lambda_j, \quad (4)$$

where $\mathbb{I}$ is the indicator function and $N$ is the number of tokens in the input sequence.

For each transferred question/source text entity pair $(q, e) \in \mathcal{Q}_t$, we first use Contriever (Izacard et al., 2022) to obtain the context $c$, i.e., the top-$k$ most relevant facts to $q$ in $\mathcal{C}$ via maximum inner-product search (Shrivastava and Li, 2014). Next, the question $q$, entity $e$ (specificity guidance), and context $c$ are fed through the specificity-aware QA model $p(\langle a_i, a_j \rangle | c, q, e)$. We record the predicted answer $a = \langle a_i, a_j \rangle$ with the highest likelihood (sum of start and end likelihoods) under length $m$.

We map each unique entity $e$ in $\mathcal{Q}_t$ to the answer $a$ with the highest total likelihood. This process returns a map $\mathcal{E}$ with each source text entity $e$ as the key and its transferred entity $a$ as the value.

## 3.4 Source Text Infilling

Lastly, we infill the source text $\mathcal{D}_s$, replacing each entity $e$ with its mapped entity $a$ in $\mathcal{E}$. We describe zero-shot and supervised infilling methods below:
***Zero-shot:*** Given that each entity $e$ appears in the source text $\mathcal{D}_s$, we replace every occurrence of $e$ with $a$ and $t_s$ with $t_t$ to create the target text $\mathcal{D}_t$.
***Supervised:*** We train the LED language model (Beltagy et al., 2020) to generate the target text $\mathcal{D}_t$ using the source topic $t_s$, source text $\mathcal{D}_s$, target topic $t_t$, corpus $\mathcal{C}$, and each transferred entity $a$ (surrounded by <|answer|> tokens). This process is similar to keyword-guided text generation techniques (Mao et al., 2020; Narayan et al., 2021). Overall, the supervised version of ModQGA allows the model to have more flexibility during infilling.

Although ModQGA is designed primarily as a 0-shot text fact transfer model, using custom components trained on external SQuAD datasets, altering the infilling process allows us to fairly compare our model with supervised baselines (§4.2).

## 3.5 The ModQGA Framework

In Algorithm 1, we use the above components to design ModQGA. First, ModQGA performs end-to-end question generation with the BART model

**Algorithm 1** ModQGA

---
1: **procedure** MODQGA($\mathcal{D}_s, t_s, t_t, \mathcal{C}, n, m, k$)
2:    $\mathcal{Q}_s \leftarrow \{\}, \mathcal{Q}_t \leftarrow \{\}, \mathcal{E} \leftarrow [map : E \rightarrow (A, S)]$
3:    **while** $|\mathcal{Q}_s| < n$ **do**
4:        $\mathcal{Q}_s \leftarrow \mathcal{Q}_s \cup$ E2E-QG($\mathcal{D}_s, t_s$)
5:        $\mathcal{Q}_t \leftarrow \mathcal{Q}_t \cup$ QUESTIONTRANSFER($\mathcal{Q}_s, t_s, t_t$)
6:    **for** $(q, e) \in \mathcal{Q}_t$ **do**
7:        $c \leftarrow$ CONTRIEVER($q, k, \mathcal{C}$)
8:        $a, score \leftarrow$ SA-QA($q, e, c, m$)
9:        $a', score' \leftarrow \mathcal{E}(e)$          ▷ Lookup $e$ in $\mathcal{E}$ map
10:       **if** $score > score'$ **then**
11:           $\mathcal{E}(e) \leftarrow (a, score)$      ▷ Update best answer
12:    $\mathcal{D}_t \leftarrow$ INFILL($\mathcal{D}_s, \mathcal{E}$)
13:    **return** $\mathcal{D}_t$

---

$p(\mathcal{Q}_s|\mathcal{D}_s, t_s)$, to generate $n$ question/entity pairs $\mathcal{Q}_s$ covering the factual content of the source text $\mathcal{D}_s$. ModQGA then transfers the questions in $\mathcal{Q}_s$ from the source topic $t_s$ to the target topic $t_t$ to create $\mathcal{Q}_t$, which has specific and generic questions. For each transferred question $q$ and source entity $e$ in $\mathcal{Q}_t$, ModQGA performs specificity-aware QA with the BERT model $p(\langle a_i, a_j \rangle | c, q, e)$. We build the map $\mathcal{E}$, containing each source entity $e$ mapped to the answer $a$ with the highest likelihood, to represent its transferred entity. Last, using $\mathcal{E}$, ModQGA infills the source text $\mathcal{D}_s$ to create the target text $\mathcal{D}_t$, either in a 0-shot or supervised manner.

## 4 Experimental Setup

We provide a detailed setup in Appendix A.

### 4.1 Datasets

We adapt the following tasks and datasets to construct parallel corpora for text fact transfer:
**1)** *Expository Text Generation* (ETG) uses topic-related sentences to create multi-sentence factual texts in a consistent style (Balepur et al., 2023). We adapt the **U.S. News** and **Medline** datasets, spanning college and medical domains. We use the output text as the target text and retrieve/create training examples for the source text (see Appendix A.1). We use the document titles for the source/target topics, and the provided corpus for $\mathcal{C}$.
**2)** *Relationship triples* have a subject $x$, predicate $y$, and relation $r$ between $x$ and $y$. We adapt the **t-REX** (Elsahar et al., 2018) and **Google** (Orr, 2013; Petroni et al., 2019) relationship triple datasets. t-REX contains **open-domain** relations, while Google contains biographical relations. The open-domain nature of t-REX allows us to assess the adaptability of each baseline. We obtain triples that share a relation $r$ (i.e., $\langle x_1, r, y_1 \rangle$ and

$\langle x_2, r, y_2 \rangle$) and use $x_1 \cdot r \cdot y_1$ as the source text and $x_2 \cdot r \cdot y_2$ as the target text ($\cdot$ denotes concatenation). We use $x_1$ and $x_2$ as the source and target topics. For t-REX, we use the Wikipedia texts in the dataset for $\mathcal{C}$, and for Google, we scrape sentences from the top-7 web pages queried with $x_2$.

### 4.2 Baselines

We compare *zero-shot* ModQGA (**0-shot ModQGA**) with the following zero-shot baselines:
**1) 0-Shot GPT:** We use a zero-shot prompt (Appendix A.3) instructing GPT-3.5 to create the target text using the source text, source topic, and target topic. This model uses its internal knowledge.
**2) 0-Shot GPT+Retr:** We add the top-5 retrieved facts from $\mathcal{C}$ as an extra input to 0-Shot GPT.
**3) SourceCopy:** We trivially copy the source text as the predicted output for the target text.

When parallel data exists in text style transfer, seq2seq models are typically used (Jin et al., 2022). Thus, for our parallel text fact transfer setting, we compare *supervised* ModQGA (**ModQGA-Sup**) with the following supervised seq2seq models:
**1)** $z$-**Shot GPT:** We construct a $z$-shot prompt for GPT-3.5 to generate the target text with the source text, source topic, and target topic as inputs. This model relies on its internal knowledge.
**2)** $z$-**Shot GPT+Retr:** We add the top-5 retrieved facts from $\mathcal{C}$ as an extra input to $z$-Shot GPT.
**3) LED:** LED (Beltagy et al., 2020) is a seq2seq LM based on the Longformer. LED produces the target text using the source text, source topic, target topic, and corpus as inputs. This model is ModQGA-Sup without the transferred entities as inputs.
**4) BART+Retr:** Similar to RAG (Lewis et al., 2020b), we retrieve the top-25 facts from $\mathcal{C}$ and train BART to generate the target text using the source text, source/target topics, and retrieved facts.

All GPT-3.5 models are `gpt-3.5-turbo` with a temperature of 0.2. Models that perform retrieval use the same Contriever setup (Izacard et al., 2022) as ModQGA. The input query used is the source text $\mathcal{D}_s$ with every occurrence of $t_s$ replaced with $t_t$. We found that this query outperforms solely the target topic $t_t$, as it provides the Contriever context as to which information to search for (see Table 6).

### 4.3 Quantitative Metrics

We measure the output similarity of the predicted and target texts with ROUGE-1/2 (**R1/R2**) and

| Dataset | Model | R1 | R2 | BLEU | Halluc | FactCC | NLI-Ent | Length |
|---|---|---|---|---|---|---|---|---|
| U.S. News | 0-shot ModQGA (**Ours**) | **0.934** | **0.890** | **0.865** | 0.29 | **0.650** | **0.708** | 1.01 |
| | 0-Shot GPT | 0.881 | 0.814 | 0.774 | 4.84 | 0.489 | 0.420 | 1.03 |
| | 0-Shot GPT+Retr | 0.832 | 0.767 | 0.679 | 3.65 | 0.534 | 0.587 | 1.16 |
| | SourceCopy | 0.795 | 0.682 | 0.671 | 0.00 | 0.220 | 0.185 | 1.00 |
| Medline | 0-shot ModQGA (**Ours**) | 0.724 | **0.605** | **0.579** | 0.00 | 0.915 | **0.502** | 0.97 |
| | 0-Shot GPT | **0.732** | 0.599 | 0.486 | 0.91 | **0.958** | 0.447 | 1.29 |
| | 0-Shot GPT+Retr | 0.476 | 0.338 | 0.176 | 0.69 | 0.825 | 0.231 | 2.60 |
| | SourceCopy | 0.559 | 0.417 | 0.400 | 0.00 | 0.890 | 0.034 | 1.00 |
| Google | 0-shot ModQGA (**Ours**) | **0.929** | **0.914** | **0.857** | 0.82 | **0.589** | **0.621** | 1.00 |
| | 0-Shot GPT | 0.838 | 0.794 | 0.670 | 3.56 | 0.245 | 0.200 | 1.01 |
| | 0-Shot GPT+Retr | 0.698 | 0.609 | 0.315 | 2.50 | 0.502 | 0.222 | 1.89 |
| | SourceCopy | 0.455 | 0.350 | 0.082 | 0.00 | 0.078 | 0.000 | 1.02 |
| t-REX | 0-shot ModQGA (**Ours**) | **0.841** | **0.781** | **0.721** | 0.58 | 0.722 | **0.609** | 1.05 |
| | 0-Shot GPT | 0.780 | 0.699 | 0.490 | 6.75 | **0.798** | 0.538 | 1.30 |
| | 0-Shot GPT+Retr | 0.585 | 0.483 | 0.157 | 3.89 | 0.739 | 0.261 | 3.15 |
| | SourceCopy | 0.497 | 0.376 | 0.350 | 0.00 | 0.004 | 0.017 | 1.00 |

Table 1: Quantitative comparison of zero-shot text fact transfer models in *output similarity* (R1, R2, BLEU) and *factuality* (Halluc, FactCC, NLI-Ent). Best results are in **bold**, barring SourceCopy Halluc, as it will always be 0.

| Dataset | Model | R1 | R2 | BLEU | Halluc | FactCC | NLI-Ent | Length |
|---|---|---|---|---|---|---|---|---|
| U.S. News | ModQGA-Sup (**Ours**) | **0.967** | **0.953** | **0.944** | **0.33** | **0.901** | **0.889** | 1.01 |
| | 3-Shot GPT | 0.883 | 0.819 | 0.787 | 5.24 | 0.357 | 0.430 | 1.03 |
| | 7-Shot GPT+Retr | 0.909 | 0.863 | 0.848 | 4.01 | 0.422 | 0.482 | 1.02 |
| | LED | 0.958 | 0.941 | 0.933 | 1.05 | 0.838 | 0.815 | 1.02 |
| | BART+Retr | 0.892 | 0.839 | 0.821 | 2.94 | 0.669 | 0.652 | 1.00 |
| Medline | ModQGA-Sup (**Ours**) | **0.870** | **0.807** | **0.785** | 0.22 | **0.976** | **0.725** | 0.99 |
| | 3-Shot GPT | 0.778 | 0.668 | 0.589 | 1.17 | 0.969 | 0.584 | 1.16 |
| | 7-Shot GPT+Retr | 0.721 | 0.606 | 0.568 | 0.49 | 0.927 | 0.460 | 1.04 |
| | LED | 0.850 | 0.780 | 0.760 | 0.30 | 0.962 | **0.725** | 0.98 |
| | BART+Retr | 0.817 | 0.732 | 0.716 | 1.03 | 0.955 | 0.605 | 1.01 |
| Google | ModQGA-Sup (**Ours**) | **0.947** | **0.937** | **0.899** | 1.52 | **0.737** | **0.714** | 1.00 |
| | 3-Shot GPT | 0.846 | 0.809 | 0.630 | 1.32 | 0.546 | 0.415 | 1.17 |
| | 10-Shot GPT+Retr | 0.812 | 0.773 | 0.614 | 4.50 | 0.541 | 0.467 | 1.14 |
| | LED | 0.938 | 0.926 | 0.878 | 1.54 | 0.683 | 0.661 | 1.00 |
| | BART+Retr | 0.943 | 0.932 | 0.890 | **1.04** | 0.732 | 0.696 | 1.00 |
| t-REX | ModQGA-Sup (**Ours**) | 0.833 | 0.761 | 0.735 | **0.65** | 0.661 | 0.539 | 0.98 |
| | 3-Shot GPT | 0.710 | 0.598 | 0.444 | 9.67 | **0.862** | 0.500 | 1.25 |
| | 10-Shot GPT+Retr | 0.742 | 0.666 | 0.499 | 5.88 | 0.591 | 0.536 | 1.29 |
| | LED | 0.816 | 0.753 | 0.720 | 0.66 | 0.670 | 0.478 | 1.03 |
| | BART+Retr | **0.883** | **0.835** | **0.812** | 0.83 | 0.835 | **0.722** | 1.01 |

Table 2: Quantitative comparison of supervised text fact transfer models in *output similarity* (R1, R2, BLEU) and *factuality* (Halluc, FactCC, NLI-Ent). Best results are in **bold**, second best results are underlined.

BLEU (Lin, 2004; Papineni et al., 2002), serving as proxies for style preservation of the source text.

To evaluate factuality, we adopt three metrics: **1) Halluc** calculates the average percentage of tokens that are extrinsically hallucinated, meaning that they do not appear in the corpus $\mathcal{C}$ or source text $\mathcal{D}_s$; **2) FactCC** (Kryscinski et al., 2020) is a classifier that predicts if any factual errors exist between a source text and claim. We use the true output as the source and each sentence of the generated text as the claim, and report the proportion of sentences with no factual errors; **3) NLI-Ent** uses textual entailment to predict whether a claim is entailed by a source (Maynez et al., 2020). We train a DistilBERT (Sanh et al., 2019) classifier on the MNLI dataset (Williams et al., 2018) (accuracy of 0.82) and report the proportion of sentences in the generated text that are entailed by the true output.

All metrics are reported from a single run.

## 5 Results

### 5.1 Quantitative Performance

In Table 1, we see that 0-shot ModQGA excels at text fact transfer, achieving the strongest results in 22/24 metrics. This is impressive given that ModQGA has significantly less parameters than GPT-3.5 (0.8B vs 175B). We also note that 0-shot ModQGA outperforms ModQGA-Sup on opendomain t-REX, showing that our 0-shot model is more adaptable than its supervised version, but is surpassed by BART+Retr, opening the door to research in 0-shot text fact transfer to close this gap.

In Table 2, we find that ModQGA-Sup outperforms baselines on three datasets (17/18 metrics on U.S. News/Medline/Google), and achieves the second strongest results on t-REX. Further, ModQGA-

| Metric | Zero-Shot | | | Supervised | | |
|---|---|---|---|---|---|---|
| | Ours | Equal | GPT | Ours | Equal | LED |
| *U.S. News*-Style | **59.0** | 28.0 | 13.0 | 3.0 | 91.0 | **6.0** |
| *U.S. News*-Fact | **47.0** | 49.0 | 4.0 | **46.0** | 49.0 | 5.0 |
| *Google*-Style | **86.0** | 11.0 | 3.0 | **6.0** | 94.0 | 0.0 |
| *Google*-Fact | **13.0** | 78.0 | 9.0 | **14.0** | 84.0 | 2.0 |

Table 3: Pairwise comparison of 0-shot ModQGA vs 0-Shot GPT+Retr and ModQGA-Sup vs LED. We evaluate with **Fact**uality/**Style** on U.S. News/Google. Results are averaged w.r.t. annotators and reported as a percent. The favored models ignoring ties (Equal) are in **bold**.

Sup surpasses LED in 23/24 metrics, meaning that our extra input of transferred entities is valuable for improving the style and factuality of seq2seq models in text fact transfer. These findings suggest that our strategy of identifying entities, transferring entities between topics, and infilling, can outperform generating text solely in a seq2seq manner.

Finally, we note that GPT-3.5 fails to produce factual text, obtaining much lower factuality scores that are not always improved by using the corpus $\mathcal{C}$. The LLM also struggles to adhere to the style of the source text, shown by the lower output similarity scores and larger length ratios. Thus, text fact transfer highlights the limitations of GPT-3.5 with preserving factuality and style, meaning that our task could benchmark these capabilities of LLMs.

## 5.2 Human Evaluation

We invite two computer science and engineering students to evaluate 50 generated outputs from U.S. News and Google on *style* (i.e. which output best matches the source text style) and *factuality* (i.e. which output is more factual). Following best practices, we use a pairwise comparative evaluation (Lewis et al., 2020b). To study the issues of 0-shot LLMs, we compare 0-shot ModQGA and 0-Shot GPT+Retr, and to study if the extra inputs of transferred entities aid seq2seq models, we compare ModQGA-Sup and LED.

In Table 3, the evaluator ratings indicate that 0-shot ModQGA better preserved style compared to 0-Shot GPT+Retr in over 55% of cases on both datasets and was more factual on U.S. News in 47% of cases, highlighting that ModQGA is a preferred choice for the challenging task of 0-shot text fact transfer. Further, evaluators indicated that ModQGA-Sup outperformed LED in factuality in 46% of cases on U.S. News, once again suggesting that our transferred entities can improve the factual

| Model | R1 | R2 | BLEU | FactCC | NLI-Ent |
|---|---|---|---|---|---|
| Full Model | **0.724** | **0.605** | **0.579** | **0.915** | **0.502** |
| No Generic | 0.680 | 0.553 | 0.527 | 0.889 | 0.215 |
| Normal QA | 0.686 | 0.562 | 0.522 | 0.825 | 0.278 |

Table 4: Ablation study for 0-shot ModQGA on Medline. No Generic removes generic questions during question transferring, and Normal QA uses a BERT-based question answering model without specificity guidance.

**Question:** What is the setting of the University of Florida?

| **Entity:** urban
**Answer:** residential | **Entity:** Tucson, Arizona
**Answer:** Tallahassee, Florida |
|---|---|
| **Entity:** Philadelphia
**Answer:** Tallahassee | **Entity:** The Nation's Capital
**Answer:** The State Capital |

| **Entity:** East
**Answer:** North | **Entity:** Oklahoma
**Answer:** Tallahassee | **Entity:** 200
**Answer:** 185 |
|---|---|---|

Figure 3: Examples of our QA model altering the specificity of its answer depending on the entity. The question was obtained by running ModQGA on U.S. News. Green/red text indicates correct/incorrect answers.

accuracy of seq2seq models. These findings parallel our quantitative results (§5.1), reinforcing that ModQGA can effectively transfer factual content without sacrificing the style of the source text.

## 5.3 Ablation Studies

We conduct an ablation study (Table 4, full results Appendix 8) and note that the use of generic questions and specificity guidance improve the output similarity and factuality of 0-shot ModQGA. We find the specificity result to be noteworthy, as it means controlling specificity can enhance the performance of 0-shot text fact transfer frameworks.

## 5.4 Specificity-Aware QA Analysis

In Figure 3, we assess the abilities of our specificity-aware QA model. Overall, we find that the model does use the specificity of the entity guidance, having the ability to provide a regional descriptor ("residential"), city ("Tallahassee"), city and state ("Tallahassee, Florida"), and city descriptor ("The State Capital"). This suggests that our model has at least some ability to control the specificity of its answers.

Despite these strengths, our QA model may still err. Specifically, the model may identify a part of the context that matches the specificity of the entity, even though it does not correctly answer the question (e.g., "North" comes from the context "North side of campus," but the correct answer is "South"). Further, the model may be biased towards answers

that match the length of the entity, even if the specificity is not matched (e.g., predicting "Tallahassee" instead of "Florida"). Finally, if the provided entity is drastically unrelated to the question (e.g., "200"), so will the answer (e.g., "185"). Controlling specificity is a difficult task (Huang et al., 2022), but we believe our specificity-aware QA model reveals a potential direction to address this problem.

## 5.5 Sample Outputs

In Appendix B.3, we present examples of target texts generated by ModQGA and other baselines.

## 6 Conclusion

We propose the task of text fact transfer and develop ModQGA to overcome the difficulty of LMs to perform our task. ModQGA leverages a novel combination of end-to-end question generation and specificity-aware question answering to perform text fact transfer. Through experiments on four datasets, including human evaluation, we find that 0-shot and supervised ModQGA excel in style preservation and factuality on a majority of datasets. We conduct an ablation study to reveal the strengths of our design choices of ModQGA. Finally, we perform a qualitative analysis of our specificity-aware question answering model, which shows at least some ability to control the specificity of its answers.

## 7 Limitations

One limitation of 0-shot ModQGA is that it has a slower inference time compared to the 0-Shot GPT models. Although our model shows improvements in factuality and style over the GPT models, we acknowledge that it is important to ensure our framework is computationally efficient. The slowest part of ModQGA is the ensembling of multiple questions during question answering. Hence, we believe future research could improve upon ModQGA by identifying a subset of generated questions that are likely to produce high-quality answers, and only using this subset in ModQGA. This could make contributions to an interesting research area of high-quality question identification.

Further, we assume that the factual corpora used in our tasks are error-free and do not contain contradictions. Hence, we did not assess how any text fact transfer framework would perform if placed in a setting with misinformation. This could be an interesting future setting for text fact transfer, as any model to solve the task would now have to incorporate the extra step of fact verification, making the task more similar to its downstream use case.

## 8 Ethics Statement

The goal of text fact transfer is to transfer the factual content of a source text while preserving its original style, which we accomplish by designing ModQGA. As mentioned in the introduction, some downstream applications of text fact transfer could include automatically generating news for current events by leveraging a previous news article for a similar event or repurposing existing educational materials for new subjects. However, as with all text generation frameworks, a model like ModQGA which is designed for text fact transfer could still hallucinate factual errors. Hence, to avoid the spread of misinformation and inaccurate factual content, ample considerations and thorough evaluations must be made before leveraging a text fact transfer framework in downstream applications.

## 9 Acknowledgements

We thank the anonymous reviewers for their feedback. This material is based upon work supported by the National Science Foundation IIS 16-19302 and IIS 16-33755, Zhejiang University ZJU Research 083650, IBM-Illinois Center for Cognitive Computing Systems Research (C3SR) - a research collaboration as part of the IBM Cognitive Horizon Network, grants from eBay and Microsoft Azure, UIUC OVCR CCIL Planning Grant 434S34, UIUC CSBS Small Grant 434C8U, and UIUC New Frontiers Initiative. Any opinions, findings, and conclusions or recommendations expressed in this publication are those of the author(s) and do not necessarily reflect the views of the funding agencies.

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

## A Experimental Setup

### A.1 Datasets

The U.S. News dataset in ETG already follows a very consistent style that can be adapted for text fact transfer. Hence, we take each output as the target text, and match it with a source text by selecting a random example from the training set. We match descriptions for public colleges with other random descriptions for public colleges, and the same for private colleges, as there are slight differences in the style between these descriptions when describing tuition. The last sentence of the public college descriptions follow the form "The in-state tuition is X; the out-of-state tuition is Y", while the last sentence of the private college descriptions follow the form "The tuition is X".

For the Medline dataset in ETG, retrieving a similar example for the source text cannot be done in the same way, as the style is less consistent. Hence, we first convert each output document to a consistent style by performing question answering using the output as the context with the following questions: 1) "What is [topic] used to treat?"; 2) "What class of medications does [topic] fall into?"; 3) "How does [topic] work?". Using these answers, we construct a document through the template: `[Topic] is used to treat [(1)]. It belongs to a class of medications called [(2)]. It works by [(3)]`. For question answering, we leverage the RoBERTa-Base model trained on SQuAD[5]. To ensure all collected outputs fall into this template, we discard documents which provide a negative logit score to any of the three questions. We then use the same process as U.S. News to match source and target texts.

The Google and t-REX datasets do not require any modifications, as we simply pair source texts and target texts by finding relation triples that share a relation. To obtain the factual corpora $\mathcal{C}$ for each target topic $t_t$ in Google, we web scrape using the query "$t_t$ Wikipeida." We keep only alphanumeric characters and punctuation, and decode the text with unidecode. We qualitatively analyzed a sample of corpora and did not find any personal identifiable information. To be safe, we use the Presidio[6] analyzer provided by Microsoft and remove all sentences with the following detected entities (prediction score > 0.3): "PHONE NUMBER", "CRYPTO", "EMAIL ADDRESS", "IBAN CODE", "IP ADDRESS", "MEDICAL LICENSE", "US BANK NUMBER", "US DRIVER LICENSE", "US ITIN", "US PASSPORT", "US SSN".

We provide summary statistics of each dataset in Table 5. All datasets are in English.

### A.2 Training Setup

The question generation model of ModQGA is trained with BART Large (406M), using a batch size of 8, learning rate of 2e-5, weight decay of 0.01, 500 warmup steps, 8 gradient accumulation steps, and 3 training epochs. The question answering model of ModQGA is trained with BERT Large (340M) using the same parameters. We select answers spans with a maximum length $m$ equal to two times the length of the entity specificity guidance. We generate $n = 10$ sequences in end-to-end question generation with nucleus decoding (top-$p = 0.75$). During retrieval, we select $k = 5$ texts.

The infilling for ModQGA-Sup and LED are implemented with the same LED model (Beltagy et al., 2020) (149M), using a batch size of 1, learning rate of 5e-5, and 1500 warmup steps. We train each model for 15 epochs and after training, load the model with the lowest validation loss with respect to each epoch. We use a maximum input size of 16384 to encode the input corpus, a maximum output length of 256 for U.S. News and Medline, and a maximum output length of 64 for Google and t-REX. The training time for this model was around 10 hours on each dataset. The BART model in BART+Retr is trained with the same parameters and similar model size (140M) as the LED model, but instead using a maximum input size of 1024. Using the same strategy, we train the model for 10 epochs and after training, load the model with the lowest validation loss with respect to each epoch. We ensured that the validation loss of each seq2seq model converged on our datasets.

All GPT-3.5 models are `gpt-3.5-turbo` (175B) with a temperature of 0.2. For U.S. News and Medline, we set the maximum output length to 256, and for Google and t-REX, we set the maximum output length to 64. The Retriever used by all baselines is the Contriever model (Izacard et al., 2022) fine-tuned on MS-MARCO (Nguyen et al., 2016), which is based on BERT (110M). The input query is the source text with every occurrence of

---

[5]https://huggingface.co/deepset/roberta-base-SQuAD2

[6]https://microsoft.github.io/presidio/analyzer/

the source topic replaced with the target topic. In Table 6, we show that this setup outperforms solely using the target topic as the query.

We retrieve $k = 25$ texts for the BART+Retr model, and $k = 5$ texts for the GPT models We found that retrieving more than $k = 5$ texts would limit the number in-context examples that we could provide to GPT-3.5, and we found that these in-context examples were essential to improve the performance of the GPT+Retr models (See Appendix A.3, which also contains the prompts used for each GPT model).

Hyperparameters were manually selected (no search) by assessing validation loss. All models were trained on a single NVIDIA A40 GPU. R1, R2, and BLEU were calculated using the hugging-face Evaluate library.[7]

### A.3 GPT Prompts and Considerations

We provide a preliminary analysis to study how prompt size affects the few-shot GPT-3.5 models for text fact transfer on the Google dataset in Table 7. Interestingly, we find that increasing the size of the in-context examples from 3 to 10 worsens the performance of the GPT-3.5 models that do not use retrieval, but also increases the performance of the GPT-3.5 models that do use retrieval. This could indicate that LLMs are highly sensitive to the in-context examples for text fact transfer.

We provide the prompt used for the 0-shot GPT-3.5 models in Figure 4 and the prompt used for the $z$-shot GPT-3.5 models in Figure 5. When creating the prompt for the 0-shot model, we tested slight variations of the prompt shown in Figure 4 on the validation sets and ultimately found the one shown to work the best.

Given the sensitivity of 0-shot GPT-3.5, we acknowledge that there likely exists a prompt that could boost the performance of this model. However, looking at Tables 1 and 2, we observe that 0-shot ModQGA consistently outperforms the $z$-shot GPT-3.5 models on all datasets except for Medline. Given this outcome and that $z$-shot GPT-3.5 is expected to outperform 0-shot GPT-3.5 regardless of the prompt, we believe that, at the very least, 0-shot ModQGA will outperform the 0-shot GPT-3.5 models across varied prompt formats on all datasets except Medline.

---

[7]https://huggingface.co/docs/evaluate/index

## B Results

### B.1 Full Ablation

We display the full ablation results in Table 8. We find that the use of a specificity aware question answering model and ensembling of generic questions consistently improve the factuality and style of 0-shot ModQGA.

### B.2 Human Evaluation

We build the human evaluation interface using PrairieLearn (West et al., 2015). Instructions given to the annotators are shown in Figure 6, and a screenshot from the interface is shown in Figure 7. The model outputs were randomized in each comparison. We use Gwet's AC2 (Gwet, 2008) to measure annotator agreement, given the presence of high agreement in our evaluation (e.g., over 90% of supervised models annotated as having equal style). We compute a value of 0.71, indicating good agreement.

### B.3 Sample Outputs

We provide examples of outputs produced by Mod-QGA (0-shot and supervised) along with their respective baselines in Tables 9, 10, 11 for U.S. News, Medline, and Google, respectively.

| Dataset | # Train/Valid/Test | Avg Output Length | Avg Corpus Size |
|---|---|---|---|
| U.S. News | 315 / 39 / 79 | 72.61 | 531.61 |
| Medline | 284 / 24 / 97 | 31.01 | 1064.91 |
| Google | 777 / 224 / 109 | 7.21 | 116.38 |
| Analogy | 496 / 68 / 115 | 6.09 | 205.24 |

Table 5: Summary statistics of text fact transfer datasets.

| Dataset | Model | R1@5 | R1@10 | R1@15 | R1@25 |
|---|---|---|---|---|---|
| U.S. News | Contriever-Source | **0.574** | **0.649** | **0.689** | **0.762** |
| | Contriever-Topic | 0.383 | 0.495 | 0.569 | 0.674 |
| Medline | Contriever-Source | **0.498** | **0.643** | **0.714** | **0.799** |
| | Contriever-Topic | 0.316 | 0.519 | 0.603 | 0.739 |
| Analogy | Contriever-Source | **0.911** | **0.951** | **0.956** | **0.966** |
| | Contriever-Topic | 0.818 | 0.912 | 0.933 | 0.948 |
| Google | Contriever-Source | **0.704** | **0.717** | **0.736** | **0.797** |
| | Contriever-Topic | 0.697 | 0.711 | 0.717 | 0.795 |

Table 6: Baseline query comparison with Contriever using average ROUGE-1 recall compared to the ground truth output on the validation set. ROUGE-1 recall will allow us to assess what proportion of the tokens from the output are covered by the information retrieved using Contriever (Note: This does not include the tokens covered by the source text). $k$ denotes the number of facts retrieved from the factual corpus. Contriever-Source uses the source text with every occurrence of the source topic replaced with target topic, while Contriever-Topic solely uses the target topic as a query. Best results are in bold.

| Model Type | Model | R1 | R2 | BLEU | Halluc | FactCC | NLI-Ent | Length |
|---|---|---|---|---|---|---|---|---|
| GPT No Retr | 3-Shot | **0.846** | **0.808** | 0.630 | **1.32** | **0.546** | **0.415** | 1.17 |
| | 10-Shot | 0.838 | 0.792 | **0.675** | 6.77 | 0.262 | 0.244 | 1.05 |
| GPT+Retr | 3-Shot | 0.628 | 0.574 | 0.382 | 11.74 | 0.508 | 0.308 | 1.35 |
| | 10-Shot | **0.812** | **0.773** | **0.614** | **4.49** | **0.541** | **0.467** | 1.14 |

Table 7: Performance analysis with respect to the number of few-shot prompts for GPT-3.5 models on the Google dataset.

| Dataset | Model | R1 | R2 | BLEU | FactCC | NLI-Ent | Length |
|---|---|---|---|---|---|---|---|
| U.S. News | Full ModQGA | **0.934** | **0.890** | **0.865** | **0.650** | **0.708** | 1.01 |
| | No Generic | 0.867 | 0.803 | 0.772 | 0.428 | 0.584 | 1.03 |
| | Normal QA | 0.883 | 0.811 | 0.760 | 0.444 | 0.639 | 1.05 |
| Medline | Full ModQGA | **0.724** | **0.605** | **0.579** | **0.915** | **0.502** | 0.97 |
| | No Generic | 0.680 | 0.553 | 0.527 | 0.889 | 0.215 | 0.97 |
| | Normal QA | 0.686 | 0.562 | 0.522 | 0.825 | 0.278 | 1.07 |
| Google | Full ModQGA | **0.929** | **0.914** | **0.857** | 0.589 | **0.621** | 1.00 |
| | No Generic | 0.925 | 0.910 | 0.843 | 0.600 | 0.613 | 1.02 |
| | Normal QA | 0.915 | 0.891 | 0.784 | **0.611** | 0.585 | 1.07 |
| t-REX | Full ModQGA | **0.841** | **0.781** | **0.721** | **0.722** | **0.609** | 1.05 |
| | No Generic | 0.769 | 0.712 | 0.689 | 0.443 | 0.113 | 1.02 |
| | Normal QA | 0.805 | 0.726 | 0.629 | 0.698 | 0.509 | 1.14 |

Table 8: Ablation comparison with output similarity metrics (R1, R2, BLEU) and factuality metrics (FactCC, NLI-Ent) for the 0-shot ModQGA models. No Generic removes generic questions during question transferring, and Normal QA uses a BERT-based question answering model without specificity guidance.

```
Information about university of nevada, las vegas: {University of Nevada,
Las Vegas is a public institution that was founded in 1957. University of
Nevada, Las Vegas' ranking in the 2022-2023 edition of Best Colleges is
National Universities, #285. With its innovative frontier spirit,
University of Nevada Las Vegas UNLV is a thriving urban research
institution with a diverse enrollment of more than 24,700 students, 4200
graduate students, 1507 faculty members, and 1,185 international students
scholars. Since its first classes were held in 1957, the University of
Nevada, Las Vegas UNLV , has undergone an amazing transformation from a
dusty outpost on the south edge of town to a thriving urban research
institution. The University of Nevada, Las Vegas is a large public
university located on an urban campus in Las Vegas, Nevada.}
Template text with the topic of michigan state: {Michigan State University
is a public institution that was founded in 1855. It has a total
undergraduate enrollment of 38,574 , its setting is suburban, and the
campus size is 5,192 acres. It utilizes a semester-based academic
calendar. Michigan State University's ranking in the 2022-2023 edition of
Best Colleges is National Universities, #77. Its in-state tuition and fees
are $14,850; out-of-state tuition and fees are $40,662.}
Minimally modify the template text so it discusses the topic of university
of nevada, las vegas. Do not deviate at all from the style or length of
the template text.
Output: {
```

Figure 4: Example zero-shot prompt for 0-shot GPT-3.5+Retr on U.S. News. The 0-shot GPT-3.5 (no Retrieval) model uses the same prompt, without the information prepended in the beginning of the prompt.

```
Template Title: {michigan state}
Template: {michigan state university is a public
institution that was founded in 1855. it has a total
undergraduate enrollment of 38,574 , its setting is
suburban, and the campus size is 5,192 acres. it utilizes
a semester-based academic calendar. michigan state
university's ranking in the 2022-2023 edition of best
colleges is national universities, #77. its in-state
tuition and fees are $14,850; out-of-state tuition and
fees are $40,662.}
Title: {university of nevada, las vegas}
Source: {university of nevada, las vegas is a public
institution that was founded in 1957. university of
nevada, las vegas' ranking in the 2022-2023 edition of
best colleges is national universities, #285. with its
innovative frontier spirit, university of nevada las
vegas unlv is a thriving urban research institution with
a diverse enrollment of more than 24,700 students, 4200
graduate students, 1507 faculty members, and 1,185
international students scholars. since its first classes
were held in 1957, the university of nevada, las vegas
unlv , has undergone an amazing transformation from a
dusty outpost on the south edge of town to a thriving
urban research institution. the university of nevada, las
vegas is a large public university located on an urban
campus in las vegas, nevada.}
Output: {
```

Figure 5: Example few-shot prompt for $z$-shot GPT-3.5+Retr on U.S. News. The $z$-shot GPT-3.5 (no Retrieval) model uses the same prompt without the source label. This is the final part of the few-shot prompt, so this prompt is preceded by $z$ in-context examples following the same format.

You will read a total of 200 pairwise outputs broken up into four sections. You will be shown a predicted output from Model A and an output from Model B, as well as a true output. Based on the true output, you will be asked to select which model (A or B) produced a better output based on **style** and **factuality**. Please use the following guidelines for style and factuality:

**Style:** How similar is the style of the model output compared to the true output? Do they organize the same information in the same order, generally using the same phrasing and same level of specificity? We are not concerned whether the factual information is correct, but rather if the factual information is being described similarly.

**Factuality:** How accurate is the information conveyed in the model ouput? Are there significant factual inconsistencies or errors? You can compare the factuality of the model output and true output, but please use Google to verify factual errors if they are not obvious

The comparison will be shown in a multiple choice format, allowing you to select for each attribute if (1) Model A was better, (2) Model B was better, or (3) the models performed similarly. Once you have made your choice, you may hit "Save and Grade". If you change your mind, you may reselect an answer choice and hit "Save and Grade" once again.

Figure 6: Instructions given to human annotators for pairwise comparison evaluation.

**True Output**

University of North Florida is a public institution. It has a total undergraduate enrollment of 14,167 , its setting is urban, and the campus size is 1,300 acres. It utilizes a semester-based academic calendar. University of North Florida's ranking in the 2022-2023 edition of Best Colleges is National Universities, #263. Its in-state tuition and fees are $6,389; out-of-state tuition and fees are $20,107.

**Model A Output**

University of North Florida is a public institution that was founded in 1971. It has a total undergraduate enrollment of 14,167, its setting is urban, and the campus size is 1,300 acres. It utilizes a semester-based academic calendar. University of North Florida's ranking in the 2022-2023 edition of Best Colleges is National Universities, #263. Its in-state tuition and fees are $6,389; out-of-state tuition and fees are $20,107.

**Model B Output**

University of North Florida is a public institution that was founded in 1816. It has a total undergraduate enrollment of 14,167, its setting is urban, and the campus size is 1,300 acres. It utilizes a semester-based academic calendar. University of North Florida's ranking in the 2022-2023 edition of Best Colleges is National Universities, #263. Its in-state tuition and fees are $6,389; out-of-state tuition and fees are $20,107.

Which output is better in **style**?

○ (a) Model A is better
○ (b) A and B are the same
○ (c) Model B is better

Which output is better in **factuality**?

○ (a) Model A is better
○ (b) A and B are the same
○ (c) Model B is better

Figure 7: Interface for human annotators for pairwise comparison evaluation.

| Model | Target Text |
|---|---|
| Ground Truth | husson university is a private institution that was founded in 1898. it has a total undergraduate enrollment of 2,572 , its setting is suburban, and the campus size is 200 acres. it utilizes a semester-based academic calendar. husson university's ranking in the 2022-2023 edition of best colleges is national universities, #331-440. its tuition and fees are $21,090. |
| Source Text | regis university is a private institution that was founded in 1877. it has a total undergraduate enrollment of 2,745 , its setting is suburban, and the campus size is 90 acres. it utilizes a semester-based academic calendar. regis university's ranking in the 2022-2023 edition of best colleges is national universities, #250. its tuition and fees are $42,220. |
| 0-shot ModQGA | husson university is a private institution that was founded in 1898. it has a total undergraduate enrollment of **2,679**, its setting is suburban, and the campus size is 200 acres. it utilizes a semester-based academic calendar. husson university's ranking in the 2022-2023 edition of best colleges is national universities, #331. its tuition and fees are $21,090. |
| 0-shot GPT | husson university is a private institution that was founded in 1898. it has a total undergraduate enrollment of **2,400**, its setting is suburban, and the campus size is **208** acres. it utilizes a semester-based academic calendar. husson university's ranking in the 2022-2023 edition of best colleges is **regional universities north**, **#119**. its tuition and fees are **$19,820**. |
| 0-shot GPT+Retr | husson university is a private institution that was founded in 1898. it has a total undergraduate enrollment of **2,679**, **with 2,264 being full time and 415 being part time**. its setting is suburban, and the campus size is **208** acres. it utilizes a semester-based academic calendar. husson university's ranking in the 2022-2023 edition of best colleges is **regional universities north**, **#119**. its tuition and fees are **$19,122**. |
| ModQGA-Sup | husson university is a private institution that was founded in 1898. it has a total undergraduate enrollment of 2,572, its setting is suburban, and the campus size is 200 acres. it utilizes a semester-based academic calendar. husson university's ranking in the 2022-2023 edition of best colleges is national universities, #331-440. its tuition and fees are $21,090. |
| 3-shot GPT | husson university is a private institution that was founded in 1898. it has a total undergraduate enrollment of **2,346**, its setting is suburban, and the campus size is **208** acres. it utilizes a semester-based academic calendar. husson university's ranking in the 2022-2023 edition of best colleges is **regional universities north**, **#136-#176**. its tuition and fees are **$19,350**. |
| 3-shot GPT+Retr | husson university is a private institution that was founded in 1898. it has a total undergraduate enrollment of **2,264**, its setting is suburban, and the campus size is **208** acres. it utilizes a semester-based academic calendar. husson university's ranking in the 2022-2023 edition of best colleges is **regional universities north**, **#136-#176**. its tuition and fees are **$19,990**. |
| LED | husson university is a private institution that was founded in 1898. it has a total undergraduate enrollment of 2,572, its setting is suburban, and the campus size is 200 acres. it utilizes a semester-based academic calendar. husson university's ranking in the 2022-2023 edition of best colleges is national universities, #331-440. its tuition and fees are **$32,560**. |
| BART+Retr | husson university is a private institution that was founded in 1898. it has a total undergraduate enrollment of **2,679**, its setting is **rural**, and the campus size is **208** acres. it utilizes a semester-based academic calendar. husson university's ranking in the 2022-2023 edition of best colleges is national universities, #331-440. its tuition and fees are **$33,834**. |

Table 9: Sample target texts generated by baselines on U.S. News with source topic Regis University and target topic Husson University. **Red text** indicates text that is factually incorrect. **Blue text** indicates text that does not follow the style of the source text.

| Model | Target Text |
|---|---|
| Ground Truth | relugolix is used to treat advanced prostate cancer. it belongs to class of medications called gonadotropin-releasing hormone receptor antagonists. it works by decreasing the amount of testosterone produced by the body |
| Source Text | hydromorphone is used to treat relieve pain. it belongs to class of medications called opiate analgesics. it works by changing the way the brain and nervous system respond to pain |
| 0-shot ModQGA | relugolix is used to treat prostate cancer. it belongs to group of medications called receptor antagonists. it works by reducing the amount of testosterone |
| 0-shot GPT | relugolix is used to treat prostate cancer. **relugolix is used to treat symptoms of prostate cancer**. it belongs to class of medications called gonadotropin-releasing hormone (gnrh) antagonists. it works by changing the way the pituitary gland produces hormones **that stimulate the growth of prostate cancer cells**. |
| 0-shot GPT+Retr | relugolix is used to treat prostate cancer **and heavy menstrual bleeding associated with uterine fibroids in premenopausal women**. it belongs to the class of medications called gnrh receptor antagonists. it works by depressing testosterone levels and **providing a less burdensome therapeutic option for patients**. **possible side effects include joint or muscle pain, high blood pressure, fluid buildup in the body, hot flashes, upset stomach, and diarrhea**. |
| ModQGA-Sup | relugolix is used to treat prostate cancer. it belongs to class of medications called gnrh antagonists. it works by decreasing the amount of testosterone that the body makes. |
| 3-Shot GPT | relugolix is used to treat advanced prostate cancer. it belongs to class of medications called gonadotropin-releasing hormone (gnrh) receptor antagonists. it works by decreasing the production of testosterone**, which can slow the growth of prostate cancer cells.** |
| 7-Shot GPT+Retr | relugolix is used to treat prostate cancer **and heavy menstrual bleeding associated with uterine fibroids**. it belongs to class of medications called gonadotropin-releasing hormone receptor antagonists. it works by decreasing the production of certain hormones in the body. |
| LED | relugolix is used to treat advanced prostate cancer. it belongs to class of medications called **androgen receptor antagonists**. it works by decreasing the amount of testosterone |
| BART+Retr | relugolix is used to treat **adult** patients with advanced prostate cancer. it belongs to class of medications called gonadotropin-releasing hormone receptor antagonists. it works by decreasing the amount of androgen made in the body |

Table 10: Sample target texts generated by baselines on Medline with source topic Hydromorphone and target topic Relugolix. **Red text** indicates text that is factually incorrect. **Blue text** indicates text that does not follow the style of the source text.

| Model | Target Text |
|---|---|
| Ground Truth | dennis davis was born in manhattan. |
| Source Text | paul downes was born in devon. |
| 0-shot ModQGA | dennis davis was born in manhattan. |
| 0-shot GPT | dennis davis was born in **devon**. |
| 0-shot GPT+Retr | dennis davis was born in **[unknown location]**. |
| ModQGA-Sup | dennis davis was born in manhattan. |
| 3-shot GPT | dennis davis was born in **institute**, **west virginia**. |
| 10-shot GPT+Retr | dennis davis was born in **devon**. |
| LED | dennis davis was born in **london**. |
| BART+Retr | dennis davis was born in manhattan. |

Table 11: Sample target texts generated by baselines on Google with source topic Paul Downes and target topic Dennis Davis. **Red text** indicates text that is factually incorrect. **Blue text** indicates text that does not follow the style of the source text.