# OpenReview forum: "Text Fact Transfer"
_EMNLP/2023/Conference — EMNLP 2023 Main_

### Official Review · Reviewer_KoEs · 2023-08-03

**Soundness:** 4

**Excitement:**

3: Ambivalent: It has merits (e.g., it reports state-of-the-art results, the idea is nice), but there are key weaknesses (e.g., it describes incremental work), and it can significantly benefit from another round of revision. However, I won't object to accepting it if my co-reviewers champion it.

**Missing References:**

You may include more recent work on style transfer:
- [StoryTrans: Non-Parallel Story Author-Style Transfer with Discourse Representations and Content Enhancing](https://aclanthology.org/2023.acl-long.827) (ACL 2023)

**Paper Topic And Main Contributions:**

This paper proposes a new task named text fact transfer, which aims to modify the factual content of a source text while preserving its original style. It is the opposite task of the original style transfer. The key idea is transferring entities between topics using a minimal text alteration approach.
Furthermore, they decompose the aforementioned task into question generation and answer generation tasks, thereby achieving their objectives. On top of that, they also constructed relevant datasets.

**Reasons To Accept:**

- Proposes a new task, text fact transfer, that could enable applications like adapting past news or educational content.
- Technically sound approach, combining question generation and answering to minimally modify source text.
- Comprehensive evaluation and promising results suggesting the model better preserves style and facts compared to baselines.

**Reasons To Reject:**

- Comparison to more recent retrieval-augmented seq2seq models could be useful.
- Lack of more in-depth and systematic case studies. For example, existing GPT models can achieve transfer in various formats through in-context learning, and LLMs have better knowledgeability. What are the advantages of your model? Although the paper shows good automatic metric results, the application proposed in the article seems to require more case studies to demonstrate the effectiveness of your methods and intuitive judgments. The case in real world is more important and impressive.

**Reproducibility:**

4: Could mostly reproduce the results, but there may be some variation because of sample variance or minor variations in their interpretation of the protocol or method.

**Reviewer Confidence:**

3: Pretty sure, but there's a chance I missed something. Although I have a good feel for this area in general, I did not carefully check the paper's details, e.g., the math, experimental design, or novelty.

---

> ### Author Rebuttal · Authors · 2023-08-26
>
> Thank you for your review! We appreciate that the reviewer finds our new task to enable various applications, believes that our approach is technically sound, and notes our comprehensive evaluation and promising results.
>
> ### RE: Retrieval Augmented Models
>
> We acknowledge your concern regarding comparisons to more recent retrieval-augmented seq2seq models. We considered comparing these models, such as retrieval-augmented models that perform reranking, but felt that ModQGA could benefit from these same changes (e.g. ModQGA could also perform reranking during the question answering step to more accurately extract transferred entities). Thus, for the scope of this work, we focus on retrieval-augmented models that perform one round of retrieval followed by one round of generation. In future work, it would be interesting to explore how recent advancements in retrieval-augmented models (i.e. those developed for question answering/generation) could be integrated into our ModQGA framework.
>
> ### RE: Case Studies
>
> Thank you for your suggestions surrounding case studies. We agree that evaluating text fact transfer with in-depth case studies can highlight the unique advantages of our model. In fact, we chose many of our datasets to parallel real-world applications. For example, the U.S. News, Medline, and Google datasets are equivalent to the real-world applications of repurposing/augmenting the text of previously written college descriptions, prescription drug information, and biographical sentences. We selected the open-domain t-REX dataset to study which models have the ability to adapt to diverse real-world settings. Leveraging these datasets, we show that GPT-3.5 struggles to produce factual text and adhere to the style of the source text (line 543), highlighting that our model has advantages in factuality and stylistic consistency in these real-world applications.
>
> Further, the qualitative analysis of our specificity-aware QA model (section 5.4) was intended to highlight the unique ability of our model to control specificity in an interpretable manner, which could be very useful in real-world settings of text fact transfer when users desire information at varying levels of specificity. In the final version, we will add more details to explain how our constructed datasets and experiments are indicative of certain real-world applications.
>
> ### RE: Missing References
>
> Thank you for pointing this out. As suggested, we will reference more recent style transfer works in the final version.

---

### Official Review · Reviewer_jgwJ · 2023-08-03

**Soundness:** 4

**Excitement:**

4: Strong: This paper deepens the understanding of some phenomenon or lowers the barriers to an existing research direction.

**Paper Topic And Main Contributions:**

This paper considers the "fact transfer" problem, in which the specifics of a sentence (e.g., specific named-entities mentioned) must be replaced without altering the style of the sentence. The proposed approach involves multiple stages. The first stage is question generation, in which it is stipulated that "all entities in the source text that need to be transferred can be viewed as an answer to a question." To generate questions, a sequence-to-sequence model is fit to the SQaAD-V2 dataset to predict questions. If a generated question contains the source topic, the topic can simply be replaced by the target topic. Next, QA is done based on the generated question, where to address the specificity of the answers, the SQuAD-V2 dataset is modified to include specificity guidance in the form of skip-gram synonyms via fastText embeddings. Finally, source text infilling is done based on the generated questions and answers to produce the target text. In the supervised case, the LED language model (Beltagy et al., 2020) is used to generate the target text, similar to keyword-guided text generation.

**Reasons To Accept:**

* Although I'm not sure if the framing is entirely correct, I do think that fact transfer is an interesting problem with potential applications. Also, the focus on factuality is welcome.
* Evaluation is pretty exhaustive, including both zero-shot and supervised settings, with carefully chosen baselines (including LLMs).
* Overall, the writing is effective and the technical details clearly laid out.
* In the discussion of limitations, the authors may have been unfair to themselves, comparing inference time of their pipeline with that of GPT3.5. presumably via the OpenAI API. My sense is that the proposed approach would actually be more computationally efficient than an LLM.

**Reasons To Reject:**

* The motivation for the task could be stronger. For example, it doesn't seem quite right to start by saying that text *style* transfer has "drawbacks," when you're considering an entirely different problem that has nothing to do with style.

* As a high-level concern, it's not obvious that meaning and style can be completely disentangled. For example, in the cited motivating application of repurposing educational materials to new subjects, it seems that certain new subjects would require not only modifying the named-entities, but also significantly altering the sentence structure to express the new subjects.

* The interesting possibility of using the proposed approach for data augmentation is mentioned in the introduction. I would have liked to see this borne out in experiments on this downstream application of the proposed framework.


**Reproducibility:**

4: Could mostly reproduce the results, but there may be some variation because of sample variance or minor variations in their interpretation of the protocol or method.

**Reviewer Confidence:**

4: Quite sure. I tried to check the important points carefully. It's unlikely, though conceivable, that I missed something that should affect my ratings.

---

> ### Author Rebuttal · Authors · 2023-08-26
>
> Thank you for your thorough review! We are delighted to hear that the reviewer finds text fact transfer to be an interesting problem with potential applications, believes our evaluation to be exhaustive and careful, and feels our writing is effective and clear. We also appreciate the reviewer’s defense of the inference time of our model compared to LLMs.
>
> ### RE: Task Motivation
>
> Thank you for your suggestions regarding our framing of the motivation for text fact transfer. We agree that the use of the word "drawbacks" in relation to text style transfer is not appropriate, since the fact that text style transfer does not cover our potential applications (e.g. educational repurposing) is not a shortcoming of the task. As suggested, we will make the motivation of text fact transfer stronger so it instead speaks to the positive benefits and unique challenges of our new task.
>
> ### RE: Meaning and Style Disentanglement
>
> We acknowledge your concern that “it's not obvious that meaning and style can be completely disentangled.” We agree and believe that depending on the application, the definition of style must be modified to ensure that meaning and style can be disentangled. For example, in our setting, we define style to be the phrasing, organization, and specificity of the text. The inclusion of phrasing allows us to measure style preservation using token overlap metrics, which can effectively calculate the similarity of the phrasing of the predicted text and ground truth output (line 501). But as you mentioned, when repurposing certain types of educational materials, the definition of style may not include the phrasing of the text, since different subjects may require different phrasings to accurately convey their concepts. Thus, the definition of style would need to be modified to exclude phrasing, but the task could still focus on preserving style in terms of organization, specificity, simplicity, engagingness, formality, or other stylistic attributes (Jin et al., 2022).
>
> Overall, the definition of style we use in the paper is appropriate for tasks such as adapting/augmenting the texts of college descriptions, medical drug information, and biographical facts, where phrasing can stay consistent. However, for the most challenging applications of text fact transfer, such as repurposing educational materials, the definition of style would potentially need to be modified to exclude phrasing. In the final version, we can add these details and clearly explain which applications of text fact transfer our current definition of style covers, and how the definition of style could be modified for more challenging applications.
>
>
> ### RE: Data Augmentation
> We completely agree that data augmentation is a very interesting possibility for text fact transfer. For the scope of this work, we focus on developing a model that can effectively perform text fact transfer and thoroughly compare our model with other baselines in diverse domains. While our primary focus was on our model's ability to preserve style and factuality in several domains, we felt it was important to also recognize its applicability to data augmentation in the introduction. In the final version, we will revise the introduction to make clear which applications we focus on in the paper, compared to what we believe are more applications that can be explored in future work.

---

### Official Review · Reviewer_vEhb · 2023-08-05

**Soundness:** 4

**Excitement:**

4: Strong: This paper deepens the understanding of some phenomenon or lowers the barriers to an existing research direction.

**Paper Topic And Main Contributions:**

The paper introduces the novel task of text fact transfer, the goal of which is to replace the factual content based on a source entity in the same style. Contrary to text style transfer, the style has to be preserved while adapting the facts. The authors propose a model which  uses question generation to generate a question related to the source entity, replaces the source entity with target entity and employs a question answering model to generate the answer corresponding to the target entity. This is then replaced in the correct position in the source text. They implement a zeroshot and a supervised version of the model. They should that their model performs considerably better than other baselines.

**Questions For The Authors:**

- Do you think that zero-shot LLM can work better on the task if you carefully design the prompts? Did you evaluate the same before choosing the prompts?
- Do you have any intuition why the factuality score is really high in some datasets compared to others?

**Reasons To Accept:**

- They introduce a new and interesting task called "text fact transfer," which has promising applications in repurposing and adapting texts and data augmentation. It also has great potential for further exploration.
- The proposed ensemble model is very intuitive and authors demonstrate that it produces competitive results in comparison to other baseline models.
- They also make available four datasets for text fact transfer which can be useful for future studies on this task.


**Reasons To Reject:**

- The zero-shot GPT 3.5 model used for comparison is known to be sensitive to the input prompt, which is not taken into account or discussed in the paper. Appendix does include a comparison of prompt length, but on general design of the prompt.
- The evaluation metrics like BLUE and ROUGE naturally is in favor of the model given that it conducts localized replacements. The factuality metric seems to be questionable looking at the score of  **Medline-SourceCopy** in Table1.

**Reproducibility:**

3: Could reproduce the results with some difficulty. The settings of parameters are underspecified or subjectively determined; the training/evaluation data are not widely available.

**Reviewer Confidence:**

3: Pretty sure, but there's a chance I missed something. Although I have a good feel for this area in general, I did not carefully check the paper's details, e.g., the math, experimental design, or novelty.

**Typos Grammar Style And Presentation Improvements:**

- Corrections: In Table 2, the second best score for FactCC should be **t-REX, BART+Retr** and not **t-REX, LED**. Underline for second best is missing in **t-REX, Halluc**

---

> ### Author Rebuttal · Authors · 2023-08-26
>
> Thank you for your review and insightful comments! We appreciate that the reviewer finds our task to be new, interesting, with promising applications, and great potential for further exploration. Further, it was encouraging to hear that the reviewer finds our model to be intuitive and producing competitive results, and that our datasets could be useful for future studies.
>
> ### RE: Zero-Shot GPT
>
> We appreciate your concern regarding the sensitivity of 0-shot GPT. When creating the prompt for this model, we tested slight variations of the prompt shown in the paper on the validation sets and ultimately found the one shown to work the best.
>
> We acknowledge that it is possible that there exists a prompt that could boost the performance of 0-shot GPT. However, looking at Tables 1 and 2, we observe that 0-shot ModQGA consistently outperforms the n-shot GPT models on all datasets except for Medline. Given this outcome and that n-shot GPT is expected to outperform 0-shot GPT regardless of the prompt, we believe that, at the very least, 0-shot ModQGA will outperform the 0-shot GPT models across varied prompt formats on all datasets except Medline. We can add these details in the final version.
>
> ### RE: Evaluation Metrics
>
> Thank you for your feedback regarding our choice of evaluation metrics.
>
> #### *ROUGE/BLEU*
>
> We believe that our use of localized replacements through question generation and answering does not automatically grant our model an unfair advantage with BLEU and ROUGE. In fact, using localized replacements is still a difficult approach, as even a minor misidentification of which entities need to be transferred in the source text can lead to much lower scores.
>
> Further, our decision to use ROUGE and BLEU was because they were fundamentally appropriate for measuring style preservation (instead of favoring any particular model). Part of our definition of style is “how the factual content is phrased” (line 51). Thus, we found it most relevant to evaluate style preservation using token overlap metrics like ROUGE and BLEU, which can effectively calculate the similarity of the phrasing of the predicted text and ground truth output. These metrics are also widely accepted in text style transfer (Jin et al., 2022), so we felt it was appropriate and best practice to use them for our task.
>
>
> #### *Factuality*
>
> Evaluating factuality is a difficult task, and there is no single metric that can capture all factual errors. Thus, we adopted a combination of factuality metrics that are designed to detect different errors. For example, FactCC is trained to detect simple factual errors such as numerical inconsistencies and entity mismatches, while NLI-Ent captures entailment relations between two pieces of text which often rely on logical inference (section 4.3). Your observation of the high FactCC score for SourceCopy-Medline (0.890) is valid and suggests that the factual inconsistencies in this dataset are not simple numerical/entity errors. However, the NLI-Ent score for SourceCopy-Medline (0.034) accurately conveys the low factuality of the SourceCopy baseline, illustrating the necessity of a multi-metric approach to factuality. We will add these details to the final version.
>
> Although we chose the automated metrics that were best fit for the task of text fact transfer, we acknowledge that they are not a perfect solution for comparing baselines. For this reason, we additionally conducted a human evaluation study (section 5.2). We find that 0-shot ModQGA outperforms 0-shot GPT+Retr and ModQGA-Sup outperforms LED in style and factuality on the two evaluated datasets, which parallel our quantitative results using automated metrics (line 573).
>
> ### RE: Presentation Improvements
>
> Thank you for pointing this out, we will make these corrections in the final version. We greatly appreciate the attention to detail!

---

### Meta-Review · Area_Chair_LScN · 2023-09-18

**Recommendation:** 5

**Metareview:**

This paper seems to introduce a new and interesting task on text fact transfer with promising applications. The proposed model seems intuitive and technically sound. The evaluation seems exhaustive with multiple real world datasets and a reasonable set of metrics. The results are competitive with clear writing. All reviewers find this paper to be mostly technically sound with moderately high excitement. Reviewers shared some concerns on sensitivity to slight variations of prompt, choice of factuality metrics, discussion on applications where style and meaning can be disentangled reasonably, and also discussion on case studies to further illustrate the effectiveness of the methods. Reviewers were mostly happy with the author's responses. Authors should include these changes in the revision, which could significantly increase the impact of this work.

---

### Decision · Program_Chairs · 2023-10-07

**Decision:**

Accept-Main

**Comment:**

This paper seems to introduce a new and interesting task on text fact transfer with promising applications. The proposed model seems intuitive and technically sound. The evaluation seems exhaustive with multiple real world datasets and a reasonable set of metrics. The results are competitive with clear writing. All reviewers find this paper to be mostly technically sound with moderately high excitement. Reviewers shared some concerns on sensitivity to slight variations of prompt, choice of factuality metrics, discussion on applications where style and meaning can be disentangled reasonably, and also discussion on case studies to further illustrate the effectiveness of the methods. Reviewers were mostly happy with the author's responses. Authors should include these changes in the revision, which could significantly increase the impact of this work.